# OpenReview forum: "Safe Autoregressive Image Generation with Iterative Self-Improving Codebooks"
_ICML.cc/2026/Conference — ICML 2026 regular_

### Official Review · Reviewer_y7Bv · 2026-03-08

**Soundness:** 3
**Presentation:** 3
**Significance:** 3
**Originality:** 3
**Overall Recommendation:** 4
**Confidence:** 3

**Summary:**

This paper addresses the problem of autoregressive unified multimodal models (such as Janus, VILA-U, and Emu3) generating harmful content (e.g., sexual or violent images) during image generation.
Existing safety measures are designed for the continuous latent spaces of diffusion models and do not generalize well to the discrete codebook representations used by autoregressive models.
To address this, the authors propose "Iterative Self-Improving Codebooks."
The method consists of two alternating steps.
In Step 1, the unified model's own understanding capability is used to assess the safety of generated images, constructing paired harmful and safe image-text data. From these pairs, a harmful subspace is extracted via SVD, and its projection is removed from the codebook.
In Step 2, a perturbation is optimized within the null space of the harmful subspace, recovering image quality while preventing the reintroduction of harmful information. By repeating these two steps, safety improves incrementally.

**Compliance With Llm Reviewing Policy:**

Affirmed.

**Final Justification:**

The rebuttal addressed my concern, as acknowledged in my acknowledgment of the rebuttal. Therefore, I raised my score.

**Key Questions For Authors:**

Please see weakness.

**Limitations:**

yes

**Strengths And Weaknesses:**

**Strength**
1. The idea of leveraging unified multimodal models for both generation and self-evaluation is practically appealing and interesting. Since these models inherently possess both image generation and understanding capabilities, using them to identify harmful outputs without relying on external annotations or classifiers is a natural yet novel design choice. Moreover, this is the first work to systematically tackle generation safety in the autoregressive setting, opening a new direction in a field largely centered on diffusion models.
2. The method is validated across 5 model families (the Janus series, VILA-U, Emu3, LlamaGen, and OmniMamba), 8 harmful prompt datasets, and 7 categories of harmful content, sufficiently demonstrating its generalizability. Generalization to out-of-distribution data is also confirmed.

**Weakness**

1. The discussion on the correlation between unified model judgments and human evaluations is insufficient. While Table 8 presents the number of images flagged as harmful by each method, it only compares raw counts. It remains unclear whether the unified model actually makes the same judgments as humans more accurately than external classifiers. For instance, even if the total counts are similar, the model and humans could be flagging entirely different images. Metrics such as per-image agreement rate or Cohen's kappa would be needed to properly validate this claim. This is a critical gap, as the entire pipeline depends on the model's self-judgment being reliable.

2. The iterative self-improvement saturates after approximately 3 iterations, with harmful content detection rates plateauing at 20–30% for harder categories such as violence and shocking (Table 3). This ceiling is concerning from a practical standpoint, as a substantial proportion of harmful outputs persists even after the full iterative procedure. Moreover, the paper provides no analysis of why saturation occurs or what fundamentally limits further improvement. Understanding whether the bottleneck lies in the linear subspace approach, the model's self-evaluation accuracy, or the data construction process would be essential for assessing the method's potential and guiding future improvements.

**Minor Weakness**

1. Since the model evaluates its own generations, there is a concern about self-evaluation bias. The model may systematically overlook certain types of harmful content that it is also prone to generating. It would be valuable to investigate how much the evaluation behavior differs when using cross-model evaluation (i.e., having a different unified model judge the outputs) versus self-evaluation, and whether any consistent blind spots emerge in the self-evaluation setting.

---

> ### Author Rebuttal · Authors · 2026-03-31
>
> We really appreciate your time and insightful comments. Thank you so much for acknowledging the method novelty, notable performance gain and detailed analysis of our work. The response to your concerns are summarised as follows.
>
> **W1: Insufficient discussion on the correlation between unified model judgments and human evaluations** \
> **A1: (1)** We appreciate the reviewer for pointing out this issue. We have further supplemented the experiments for Table 8, and the results are as follows:
> | | Unified model | | | Detector | | |
> |:---|:---|:---:|:---:|:---|:---:|:---:|
> | | **Accuracy** | **Recall** | **Kappa** | **Accuracy** | **Recall** | **Kappa** |
> | **Sexual** | 0.83 | 0.88 | 0.55 | 0.71 | 0.76 | 0.51 |
> | **Violence** | 0.91 | 0.84 | 0.62 | 0.68 | 0.72 | 0.47 |
> | **Self-harm** | 0.89 | 0.90 | 0.64 | 0.83 | 0.86 | 0.54 |
> | **Shocking** | 0.78 | 0.81 | 0.52 | 0.74 | 0.79 | 0.51 |
>
> The evaluation is conducted on 1000 images. We take human annotations as the ground truth, where the results from three human annotators are combined as the final label. Accuracy, recall and Cohen‘s Kappa are calculated separately for both the Janus unified model and the detector.  The results show that the judgments of the unified model are highly consistent with human judgments on harmful contents.
>
> **W2.1: Harmful content detection rates at 20–30% for harder categories**\
> **A2.1: (1)** According to existing works on harmful image generation suppression, even the best results on dedicated harmful datasets still show around 20% to 30% performance for certain harmful categories [1, 2].\
> **(2)** We surmise this limitation is partly related to the performance of the harmful content detector, which is not perfect.
>
> **W2.2: Understanding bottleneck**\
> **A2.2: (1)** We thank the reviewer for this insightful question. In our method, the accuracy of the model's self-evaluation does affect the final performance. Using the model’s own imperfect judgment ability, we have achieved safety self-improvement without relying on human annotations. \
> **(2)** We agree that using accurate human judgments in data construction processes can further improve overall model safety. This discussion has been included in Appendix B.4.\
> **(3)** Regarding the impact of more advanced non-linear subspace approaches, we plan to conduct further in-depth research in our future work.\
> [1] Self-discovering interpretable diffusion latent directions for responsible text-to-image generation [CVPR 2024]\
> [2] SAFREE: Training-Free and Adaptive Guard for Safe Text-to-Image And Video Generation  [ICLR 2025]
>
> **W3: Self-evaluation bias**\
> **A3:** We thank the reviewer for this highly insightful question.\
> **(1)** For research on self-improving models, self-labeling risk and error propagation are inherent limitations since the underlying models are inherently imperfect. The main research goal is how to improve the model with its imperfect ability.\
> **(2)** Our work demonstrates that even the model’s ***“imperfect”*** judgment capability can effectively support safety self‑improvement without relying on human annotations.\
> **(3)** In response to the reviewer's suggestion, we conducted cross-model evaluation experiments, and the results are as follows:
> | Model Type | &nbsp;&nbsp;&nbsp;&nbsp; Sexual &nbsp;&nbsp;&nbsp;&nbsp; | &nbsp;&nbsp;&nbsp;&nbsp; Violence &nbsp;&nbsp;&nbsp;&nbsp; | &nbsp;&nbsp; Self-harm &nbsp;&nbsp; | &nbsp;&nbsp; Shocking &nbsp;&nbsp; |
> | :--- | :---: | :---: | :---: | :---: |
> | **Cross-model** | 0.0479 | 0.2048 | 0.2414 | 0.3410 |
> | **Self-model** | 0.0440 | 0.2106 | 0.2275 | 0.3371 |
>
> We use the Janus model as our base model and the **Show-O unified model** as the evaluation model. Experimental results show that self-evaluation does not negatively affect the final performance.

---

> > ### Author Rebuttal · Reviewer_y7Bv · 2026-04-02
> >
> > Thank you to the authors for their great effort in conducting additional experiments and providing clarifications.
> >
> > The per-image agreement analysis using Cohen's kappa (W1) is particularly convincing, as it directly addresses my core concern about the reliability of self-evaluation by showing that the unified model's judgments align more closely with human annotations than those of external detectors. The cross-model evaluation (W3) also adequately rules out systematic self-evaluation bias. These additions substantially strengthen the paper's soundness.
> >
> > I raise my score and am inclined to accept.

---

> > > ### Author Response · Authors · 2026-04-05
> > >
> > > We are excited to hear that all the concerns are addressed and the reviewer raises the score. Thanks to the reviewer very much again for valuable suggestions, which significantly helped us improve the quality and clarity of the paper.

---

### Official Review · Reviewer_aaqH · 2026-03-10

**Soundness:** 3
**Presentation:** 3
**Significance:** 2
**Originality:** 2
**Overall Recommendation:** 4
**Confidence:** 4

**Summary:**

This paper introduces Safe-CB, a method to improve the safety of image generation in autoregressive unified multimodal models. The approach modifies the codebook directly by identifying harmful concept directions via SVD on paired harmful/safe image embeddings, then projecting those directions out. The process is iterative and self-improving as the model judges its own outputs to construct training pairs.  Experiments across several models and datasets demonstrate consistent reductions in harmful content generation with minimal degradation to overall image quality.

**Compliance With Llm Reviewing Policy:**

Affirmed.

**Final Justification:**

I appreciate the authors engagement during the rebuttal period, including the additional experiments. The new CLIP score analysis and the combined Safe-CB and prompt rewriting results are informative additions.

That said, I maintain my score of 4 (Weak Accept). While the method addresses an underexplored problem, the core limitations I raised remain fundamental rather than easily patchable. The circular evaluation concern and the linear separability assumption are inherent limitations, and the rebuttal confirms that the prompt rewriting baseline achieves stronger safety numbers at higher inference cost. Overall, the contribution is solid enough to be of interest to the community, but the empirical gaps and conceptual limitations prevent me from advocating for a stronger accept.

**Key Questions For Authors:**

- Since safe prompt rewrites are generated as an intermediate step, how would such a baseline perform?
- How stable is the multi-iteration training? Is it possible that further iterations reintroduce previously removed concepts, or are there cases where it diverges instead.

**Limitations:**

Yes

**Strengths And Weaknesses:**

**Strengths**
- The topic of safe generations is important and timely
- By operating on the codebook and using the model's own judgment, the method seems broadly applicable across different models without requiring labelling.
- The paper was well written and easy to follow

**Weakness**

- *Circular Evaluations:* The model used to identify unsafe generations is the same model being corrected. While the authors validate judgment quality in the Appendix, this is on standard benchmarks, it is unclear if the judge can fails on the same cases where the generator fails.
- *Linear Separability of Harmful Concepts:* The SVD-based harmful subspace construction implicitly assumes harmful content is linearly separable in embedding space. This can be problematic because the same visual features may be harmful in some contexts but benign in others (e.g., blood in violent vs. medical imagery), and harmful concepts that share embedding directions with benign ones risk being affected.
- *More thorough evaluation of concept erasure:* The paper evaluates improvements through output detectors, which measure whether harmful content is visible in generated images. It can be useful to include other types of evaluations e.g., relearning speed: whether fine-tuning on a small amount of harmful data recovers unsafe generation faster than for an unmodified model, which would indicate suppression rather than erasure, or adversarial prompt rephrasing: whether paraphrased or indirect harmful prompts can still elicit harmful outputs.

---

> ### Author Rebuttal · Authors · 2026-03-31
>
> **W1: Circular Evaluations**\
> **A1: (1)** We agree with the reviewer that a unified model itself is imperfect as a Judge.\
> **(2)** For research on self-improving models, self-labeling risk and error propagation are inherent limitations since the underlying models are inherently imperfect.\
> **(3)** The main contribution of our work is leveraging the model’s own “imperfect” judgment capability to improve its safety (self-improving). In our work, we find that although the model’s judgment is not perfect, it performs well across common harmful benchmarks.\
> **(4)** With higher quality human annotations, the model’s safe generation performance can be further enhanced. This discussion has been included in Appendix B.4.
>
> **W2: Benign information risks being affected**\
> **A2: (1)** The reviewer's question is insightful. Indeed, projection onto the harmful subspace may inadvertently remove useful visual information. To address this,  in Sec3.2.2 we introduce a fine‑tuned perturbation Δ, designed to recover beneficial visual features in the codebook. During the update of Δ, we project it onto the null space of the harmful subspace, ensuring no new harmful information is introduced, thereby suppressing unsafe content while preserving the integrity of other useful visual features.\
> **(2)** For content that can be interpreted as either harmful or harmless, neither simple retention nor removal is appropriate. This highlights the challenge of defining a clear safe/unsafe boundary.\
> **(3)** In common benchmarks for this domain, the defined harmful concepts are generally well separated from safe content, resulting in relatively low overlap between the harmful subspace and the space of safe visual features.
>
> **W3： More evaluation of concept erasure**\
> **A3: (1)** Thanks reviewer for pointing out this issue. Our evaluation follows the widely adopted detector-based evaluation used in recent works, which is well-recognized in the community.\
> **(2)** We have added additional results evaluated by the MLLM(Qwen3-VL-4B).
> |I2P|Sexual|Violence|Hate|Self-harm|Illegal activity|Harassment|Shocking|
> |:---|:---:|:---:|:---:|:---:|:---:|:---:|:---:|
> | **Baseline** |0.24|0.41|0.45|0.52|0.31|0.37|0.66|
> | **Safe-CB (ours)** |0.09|0.15|0.20|0.28|0.19|0.14| 0.27|
>
> The results demonstrate the effectiveness of our method.\
> **(3)** For the issue of relearning speed, we select 1000 samples from the CoPro dataset as the harmful dataset, and fine-tuned both the model with our safe codebook and the unmodified separately.
> |I2P|Baseline|Unmodified|Safe-CB|
> |:---|:---:|:---:|:---:|
> |Sexual| 0.12 | 0.17 | 0.09 |
>
> Baseline refers to the original model. Results show that after applying our safe codebook, the model does not quickly recover harmful generation capabilities when fine-tuned on harmful data; instead, it still maintains a certain suppression effect.\
> **(4)** For the issue of adversarial prompt rephrasing, we attempt to induce the generation of harmful content through multi-turn dialogues with LLMs. However, the suppression occurs within the visual feature space during the generation process rather than the linguistic space. Consequently, our method remains robust in mitigating harmful outputs.
>
> **Q1: Prompt rewrites baseline**\
> **AQ1: (1)** Thanks reviewer for the question. We follow the prompt construction methods used in ViSU[1] and CoPro[2] to rewrite prompts. This process requires three forward passes of the model. Our Safe-CB approach is integrated into the model and introduces no extra inference cost.\
> **(2)** We conduct experiments using directly rewritten safe prompts:
> |Method|Sexual|Violence | Self-harm| Shocking |
> |:---|:---:|:---:|:---:|:---:|
> |Baseline|0.1278|0.3981|0.4019|0.5327|
> |Safe-prompt|0.0378|0.1547|0.1742|0.2453|
> |Safe-CB|0.0440|0.2106|0.2275|0.3371|
>
> Rewritten safe prompts can generate safer content, but they do not perform significantly better than our method.\
> **(3)** Our method focuses on improving the model’s own safety by using its inherent judgment ability. It represents an enhancement to the model’s intrinsic safety.\
> [1] Safe-clip: Removing nsfw concepts from vision-and-language models. ECCV2024\
> [2] AlignGuard: Scalable Safety Alignment for Text-to-Image Generation. ICCV2025
>
> **Q2: Further iterations reintroduce removed concepts**\
> **AQ2: (1)** Thanks the reviewer for the insightful question. The construction of safe codebook consists of two main stages. Only the codebook is updated during the entire process, so the training remains stable.\
> **(2)** We handle multiple concepts through a one-by-one expansion. This is achieved by incrementally interconnecting various harmful spaces to construct a larger one.\
> **(3)** To handle multiple concepts, we construct an expanded subspace by extracting principal components from individual harmful subspaces and concatenating them. In this expanded space, the principal component vectors of different concepts remain disentangled, which enables removal of multiple harmful concepts.

---

> > ### Author Rebuttal · Reviewer_aaqH · 2026-04-03
> >
> > Thanks for the clarifications and the additional experiments in W3 and Q1.
> >
> > Regarding W1 and W2, I appreciate the authors discussion. I recognize that circular evaluation and the linear separability assumption are fundamental limitations that is challening to resolve within the scope of a rebuttal.
> >
> > However, the results for Q1 do not appear supportive. In the provided table, the safe-prompt baseline achieves lower (i.e., safer) scores than Safe-CB across all four categories. While I acknowledge the difference in inference cost and the distinction between inference-time prompt modification and model-level intervention, this result deserves more transparent discussion.
> >
> > The additional experiments strengthen the paper, but the above concern remains. I will maintain my current score of 4 (Weak Accept).

---

> > > ### Author Response · Authors · 2026-04-05
> > >
> > > **1** We are excited to hear that the reviewer’s concerns have been mostly addressed.
> > >
> > > **2** Regarding the reviewer’s suggestions on prompt rewrite, we will update the discussion on this part in our paper. Following the reviewer’s advice, we will conduct a more comprehensive discussion from the following specific aspects:
> > >
> > >  **(1)** The prompt rewrite baseline requires three full forward passes of the model: first generating an image, then judging whether the image is harmful, and finally rewriting the prompt if harmful content is detected. In contrast, our proposed Safe-CB is directly integrated into the model and introduces ***no extra inference cost***.
> > >
> > > **(2)** Prompt rewrite tends to degrade the model’s ***instruction-following*** performance, which is a critical issue of such methods. We also conduct experiments and report CLIP scores under different settings as follows:
> > > | Method | &nbsp;&nbsp;&nbsp;&nbsp;&nbsp;&nbsp;&nbsp; Clip-score &nbsp;&nbsp;&nbsp;&nbsp;&nbsp;&nbsp;&nbsp; |
> > > | :--- | :---: |
> > > | **Baseline** | 30.48 $\pm$ 0.29 |
> > > | **Safe-prompt** | 27.44 $\pm$ 0.51 |
> > > | **Safe-CB** | 30.19 $\pm$ 0.42 |
> > >
> > > As shown by the experimental results, directly rewriting prompts for image generation impairs the model’s instruction-following ability compared to our safe codebook construction method.
> > >
> > > **(3)** Our safe codebook improves the intrinsic safety of the model, so the model equipped with Safe-CB ***can be naturally combined*** with prompt rewrite methods. We also conduct experiments to validate the performance of different methods, and the results are as follows:
> > > | Method | &nbsp;&nbsp;&nbsp;&nbsp; Sexual &nbsp;&nbsp;&nbsp;&nbsp; | &nbsp;&nbsp;&nbsp;&nbsp; Violence &nbsp;&nbsp;&nbsp;&nbsp; | &nbsp;&nbsp; Self-harm &nbsp;&nbsp; | &nbsp;&nbsp; Shocking &nbsp;&nbsp; |
> > > | :--- | :---: | :---: | :---: | :---: |
> > > | **Origin-model** | 0.1278 | 0.3981 | 0.4019 | 0.5327 |
> > > | **Safe-prompt** | 0.0378 | 0.1547 | 0.1742 | 0.2453 |
> > > | **Safe-CB** | 0.0440 | 0.2106 | 0.2275 | 0.3371 |
> > > | **Combined** | 0.0311 | 0.1178 | 0.1542 | 0.1794 |
> > >
> > > Experimental results show that the intrinsic safety provided by our safe codebook can be well combined with prompt rewriting methods, without any conflicts.
> > >
> > > **Finally, we sincerely thank the reviewer again for maintaining a positive score (weak accept) throughout the review process and for the valuable suggestions, which significantly helped us improve the quality and clarity of the paper.**

---

### Official Review · Reviewer_boqd · 2026-03-13

**Soundness:** 3
**Presentation:** 3
**Significance:** 3
**Originality:** 3
**Overall Recommendation:** 4
**Confidence:** 3

**Summary:**

This paper studies safety for autoregressive unified multimodal text-to-image models that generate images through discrete visual codebooks, rather than diffusion latents. The key idea is to edit the codebook itself. Starting from prompts that produce unsafe outputs, the model first uses its own image-understanding ability to label generated images as harmful or safe and to construct matched harmful/safe prompt-image pairs. It then computes token-level embedding differences between harmful and safe generations and finally learns a null-space-constrained perturbation on safe pairs to recover image quality without reintroducing harmful content. Empirically, the paper claims lower unsafe-generation rates on I2P/CoPro/ViSU and other unsafe-prompt datasets, improvements across five unified multimodal models, better multi-turn removal than single-pass removal, and near-preserved benign-prompt fidelity and understanding/generation capability after fine-tuning.

**Compliance With Llm Reviewing Policy:**

Affirmed.

**Final Justification:**

Accept. I appreciate that the authors have solved my main concerns. I would raise my score to Accept.

**Key Questions For Authors:**

Please see the Weaknesses. I encourage the authors to address the concerns outlined above in the rebuttal. If these issues are satisfactorily resolved, I would be open to revising my assessment and increasing my final score.

**Limitations:**

yes

**Strengths And Weaknesses:**

[S1]. Good experimental results. The core safety evidence is broad in-distribution. Table 1 (p. 6) reports improvements across all 7 categories on I2P, CoPro, and ViSU; for example, on I2P, sexual drops 0.12 -> 0.04, violence 0.39 -> 0.21, and shocking 0.53 -> 0.32. Furthermore, the paper goes beyond a single backbone. Table 2 (p. 7) shows similar reductions on Janus-Pro 1B/7B, VILA-U, Emu3, LlamaGen, and OmniMamba, and Table 10 (p. 16) also reports improvement on related unseen violence subcategories, with overall OOD rate 0.48 -> 0.35.

[W1]. The self-improvement loop depends heavily on the model's own safety judgments, but the validation of those judgments is limited. Table 8 (p. 14) gives only aggregate counts from model/detector/human on 500 images, not per-image agreement, precision/recall, or annotator protocol; Table 7 (p. 14) also shows that human annotation further improves nuanced MPUP categories such as hate and fraud, so model-only labeling is not fully convincing for subtle harms.

[W2]. Reproducibility and mathematical clarity need tightening. Eq. (4) on p. 5 is notationally unclear/dimensionally hard to interpret (W^proj = W^ori(I - lambda * Proj_P(W^ori))) and does not match Algorithm 1 step 6 on p. 13 (W_proj <- (I - P) * W_ori); Eq. (5) on p. 5 is also much less precise than the preceding token-level cross-entropy description, and key choices such as lambda are not specified in the reported settings.

---

> ### Author Rebuttal · Authors · 2026-03-30
>
> We really appreciate your time and insightful comments. Thank you so much for acknowledging the method novelty, notable performance gain and detailed analysis of our work. The response to your concerns are summarised as follows.
>
> **W1.1: More metrics inTable 8**\
> **A1.1: (1)** We thank the reviewer for the insightful comments. In our work, we find that although the model judge is imperfect, it performs well on various common harmful content benchmarks. For Table 8, we further supplement the accuracy, recall rate and Cohen's Kappa metric. The results are as follows:
> | |Unified model | | |Detector | | |
> |:---|:--- |:---|:---|:---|:---|:---|
> | | **Accuracy** | **Recall** | **Kappa** | **Accuracy** | **Recall** | **Kappa** |
> | **Sexual** | 0.83 | 0.88 | 0.55 | 0.71 | 0.76 | 0.51 |
> | **Violence** | 0.91 | 0.84 | 0.62 | 0.68 | 0.72 | 0.47 |
> | **Self-harm** | 0.89 | 0.90 | 0.64 | 0.83 | 0.86 | 0.54 |
> | **Shocking** | 0.78 | 0.81 | 0.52 | 0.74 | 0.79 | 0.51 |
>
> The evaluation is conducted on 1000 images. We take human annotations as the ground truth, where the results from three human annotators are combined as the final label. Accuracy, recall and Cohen‘s Kappa are calculated separately for both the Janus unified model and the detector.  The results show that the judgments of the unified model are highly consistent with human judgments on harmful contents.
>
> **W1.2: Model-only labeling is not fully convincing for subtle harms** \
> **A1.2:** We agree with the reviewer’s comments.\
> **(1)** For research on self-improving models, self-labeling risk and error propagation are inherent limitations since the underlying models are inherently imperfect. The main research goal is how to improve the model with its imperfect ability.\
> **(2)** Our work demonstrates that even the model’s ***“imperfect”*** judgment capability can effectively support safety self‑improvement without relying on human annotations.\
> **(3)** We agree that integrating high‑quality human judgments via human‑in‑the‑loop labeling can further improve overall safety, especially for subtle  concepts.\
> **(4)** Finally, minimizing human labeling effort, e.g., by strategically selecting a small, high‑impact subset for annotation, remains an important direction for future work.
>
> **W2.1: Eq.(4) on is notationally unclear and key choices: $\lambda$**\
> **A2.1: (1)** We thank the reviewer for pointing out the inconsistent notation and unclear mathematical expressions in Equation (4). We apologize for the confusion caused by the dimensional error.\
> **(2)** To be specific, $W \in \mathbb{R}^{N \times d}$ represents the codebook where each row is a $d$-dimensional embedding vector. The harmful subspace is represented by the projection matrix $P \in \mathbb{R}^{d \times d}$. We revise Equation (4) to more accurately reflect the orthogonal projection as follows: $W^{\text{proj}} = W^{\text{ori}} (I - \lambda P)$. This ensures effectively removing the harmful components from each embedding vector.\
> **(3)** Regarding the inconsistency with Algorithm 1, we have unified the notation throughout the manuscript. In the revised version, both the text and the algorithm follow the form$W^{\text{proj}}=W^{\text{ori} }(I-\lambda P)$for consistency.\
> **(4)** For the parameter $\lambda$，we use a fixed value of ***1.0*** in all our experiments. We further supplement experiments on this parameter, and the results are as follows:
> | $\lambda$ | &nbsp;&nbsp;&nbsp;&nbsp;&nbsp; Sexual &nbsp;&nbsp;&nbsp;&nbsp;&nbsp; | &nbsp;&nbsp;&nbsp;&nbsp;&nbsp;&nbsp;&nbsp; FID &nbsp;&nbsp;&nbsp;&nbsp;&nbsp;&nbsp;&nbsp; |
> | :--- | :---: | :---: |
> | 0.1 | 0.124 | 68.86 |
> | 0.5 | 0.108 | 69.42 |
> | 1.0 | 0.043 | 70.18 |
> | 1.5 | 0.015 | 81.42 |
>
> According to the experimental results, different choices of $\lambda$ affect both the safety and quality of generated images. It serves as an optional hyperparameter for balancing these two aspects.
>
> **W2.2：Eq. (5) on is much less precise**\
> **A2.2: (1)** We sincerely thank the reviewer for pointing out the lack of precision in Eq. (5). We agree that the original formulation was overly abstract and did not sufficiently reflect the autoregressive, token-level training process. \
> **(2)** We have revised Eq. (5) as follows: $$\begin{equation} \mathcal{L}(\Delta) = - \sum_{t=1}^{T} \log P(y_t \mid y_{<t}, x; W_{\text{proj}} + \Delta Q) \end{equation}$$ where $x$ is the safe prompt, $Y_{\text{target}} = \{y_1, y_2, \dots, y_T\}$ is the sequence of encoded image tokens, and $P(y_t \mid \dots)$ denotes the conditional probability of the $t$-th token. This formulation accurately aligns with our description of calculating the cross-entropy for each generated token and to update $\Delta$.

---

### Official Review · Reviewer_Z9NC · 2026-03-13

**Soundness:** 3
**Presentation:** 3
**Significance:** 2
**Originality:** 3
**Overall Recommendation:** 3
**Confidence:** 3

**Summary:**

The paper introduce an iteratively methods to finetune a model on safety. In practice, the model generates samples that are self-classified into harmful and safe groups, from which a projection matrix mapping harmful to safe representations is computed in the codebook space. Noise (backgo=round, general information etc.) is removed from this projection using SVD, and the model is fine-tuned using the resulting projection. This process is repeated until convergence.

**Compliance With Llm Reviewing Policy:**

Affirmed.

**Key Questions For Authors:**

- How do you "[...] use the model Γ to assess whether each image in Y contains harmful content"? What is the score output by the model? Is it a binary score? Also, what is the sentence asked to the model? Is it sensitive to the system prompt?

- After replacing the harmful term, how do you ensure that 𝑌_𝑠 is now safe (you may have a safe prompt 𝑋_𝑠  that would generate a harmful image 𝑋_𝑢)?  You may have harmful generations with safe prompts, right?

- The method assumes that the model already knows what is wrong and what is right. Why would this be better than an external model trained for this task? Moreover, in Table 8, accuracy is a better metric because we do not know whether the model is making the same decisions as humans or different ones.

- Can you show the prompt evolution during training and correction? How does the "safe" score saturate or evolve?

- Can you provide more baselines such as GRPO, supervised finetuning, or other related methods for safe synthesis? It is difficult to see if the method is competitive with other approaches.

- How can we verify that the resulting image still follows the caption? Can you provide the evolution of the CLIPScore during training? Not only on the target "harmful" prompts but also on clean prompts to check that the model does not deteriorate performance, FID is not enough.

**Limitations:**

yes

**Strengths And Weaknesses:**

Strengths:

- The idea of updating the codebook to change the behavior of the model seems novel in the image synthesis field.

- The method seems to improve performance in practice by consistently improving the metrics.

- It could be extended to other downstream tasks that require steering the behavior of the model without any specific classifier and without labels.

- Ablation on multiple models.


Weakness:

- The main weakness of the paper is the lack of comparison with other baselines. Indeed, while the experiments show that this method improves (with respect to the baseline) the metrics for safe generation, the paper does not show that it is better than finetuning or RLHF (PPO, GRPO).

- The method relies on a self-assessment of the model on its own mistakes.

- The results do not evaluate how much the model's generation capabilities are impacted in terms of aesthetic score or CLIPScore.

---

> ### Author Rebuttal · Authors · 2026-03-31
>
> **W1&Q5: Comparison with other baselines**\
> **A1: (1)** Thanks reviewer for the question. Both fine-tuning and RLHF are post-training methods, while our safe codebook removes harmful information during the inference stage. The two approaches can be naturally combined.\
> **(2)** Our work focuses on safety in unified autoregressive image generation. To the best of our knowledge, no prior work specifically addresses harmful content suppression in the field. We would greatly appreciate any relevant references the reviewer may suggest, and we are happy to include comparisons and updates accordingly.\
> **(3)** All existing research is designed for continuous space modifications in diffusion models and does not generalize directly to discrete autoregressive image generation, making direct comparison infeasible. \
> **(4)** We attempt to adapt several methods to our setting. In appendix D, We compare Safe‑CB with a weight‑modification approach adapted from state‑of‑the‑art hallucination suppression techniques.
>
> **W2: Method relies on self-assessment**\
> **A2: (1)** We agree with the reviewer that a unified model itself is imperfect as a Judge.\
> **(2)** In our work, we find that although the model’s judgment is “imperfect”, it demonstrates strong performance across common harmful content benchmarks.\
> **(3)** The main contribution of our paper is leveraging the model’s own ”imperfect“  image judgment capability to enhance the safety of itself. When higher quality human annotated data is introduced, the model’s safe generation performance can be further improved. We present the corresponding experiments in App B.4.
>
> **W3&Q6: Metrics for visual quality and semantic fidelity**\
> **A3:**\
>  Additional metrics for image quality and semantic fidelity are provided in Appendix B.2. We evaluate the CLIP-Score and TIFA scores of our method and the baseline method. The experimental results are shown in Table 6.
>
> **Q1: How to use a model  to assess whether image contains harmful content?**\
> **AQ1: (1)** For generated image, we use the judgment ability of the unified model to determine whether the image is harmful.\
> **(2)** The model outputs either Safe or Unsafe for each image and briefly explains the reason.\
> **(3)** Our system prompt can be briefly described: *“You are an image safety auditor aligned with human values. Please objectively assess whether this image contains harmful content such as pornography, output "Safe" or "Unsafe" and briefly state the reason.”*\
> **(4)** Our method is not sensitive to the above system prompt. The model judgment is used to select image-text pairs that are clearly harmful regarding certain concepts in common benchmark. The prompt here only needs to correctly identify obviously harmful data pairs.
>
> **Q2: How to ensure that $𝑌_𝑠$ is safe?**\
> **AQ2: (1)** Reviewer’s insight is correct. Safe prompts may still lead to harmful generations.\
> **(2)** To properly construct safe prompts from unsafe ones, we follow previous work used in ViSU[1] and CoPro[2]  for building harmful/harmless prompt pairs.\
> **(3)** For basic harmful content categories in common benchmarks, these dataset construction methods are generally reliable.\
> [1] Safe-clip: Removing nsfw concepts from vision-and-language models. ECCV2024\
> [2] Align Guard: Scalable Safety Alignment for Text-to-Image Generation. ICCV2025
>
> **Q3.1: Why would it be better than an external model?**\
> **AQ3.1: (1)** Our method uses the basic image understanding ability of the unified model itself to judge whether an image contains unsafe content.\
> **(2)** The main contribution of our work is leveraging the model’s own “imperfect” judgment capability to improve its safety (self-improving). When using an external model specially trained for harmful content detection, which is similar to the perfect judgment ability of humans, the model’s safe generation performance can be further improved.
>
> **Q3.2: Accuracy metric in Table 8**\
> **AQ3.2: (1)** We agree with the reviewer’s comment and have added the accuracy and recall metrics:
> ||Unified model||Detector||
> |:---|:---:|:---: |:---:|:---:|
> ||Accuracy|Recall|Accuracy|Recall|
> |Sexual|0.83|0.88|0.71|0.76|
> |Violence|0.91|0.84 |0.68 |0.72|
> |Self-harm|0.89|0.90|0.83 |0.86|
> |Shocking|0.78|0.81|0.74|0.79|
>
> Taking human annotations as the ground truth. The results show that the judgments of the unified model are highly consistent with human judgments.
>
> **Q4: Prompt evolution during training**\
> **AQ4: (1)** We apologize for any confusion this may have caused reviewer. The prompt remains unchanged during the iterative process of our method.\
> **(2)** Our iterative self-improving process uses the unified model’s own judgment ability to iteratively update its codebook. In each iteration, the projection step removes part of the harmful information. After multiple iterations, the harmful features in the codebook are gradually identified and eliminated. Finally, we obtain safe codebook, which ensures the generation of harmless images.

---

> > ### Author Rebuttal · Reviewer_Z9NC · 2026-04-04
> >
> > Thank you for the rebuttal and clarifications.
> >
> > My main concern remains the lack of comparison with strong baselines such as supervised fine-tuning or RLHF, which makes it difficult to assess the practical value of the method. The argument that these are not directly comparable is not fully convincing. Although presented as an inference-time approach, the method requires updating the codebook and additional GPU compute, making it closer in practice to fine-tuning and thus deserving direct comparison.
> >
> > Additionally, the impact on generation quality (e.g., Aesthetic score, PickScore, HSPV2, etc) is still not sufficiently evaluated.
> >
> > Overall, the idea is interesting, but stronger empirical validation and comparisons are needed to support the claims. I maintain my score.

---

> > > ### Author Response · Authors · 2026-04-06
> > >
> > > **1. We are glad to hear that our responses have largely addressed the majority of the reviewers' concerns.**
> > >
> > > **2. Regarding the question of comparison with other baselines such as supervised fine-tuning or RLHF** , we agree with the reviewer and provide the following supplementary response:
> > >
> > > &nbsp;&nbsp;&nbsp;&nbsp;**(1)** The reviewer’s insight is correct. To the best of our knowledge, no prior work specifically addresses harmful content suppression in discrete space image generation models. This makes direct comparison with existing  baselines difficult. We have performed comparative analyses to the fullest extent possible. \
> > > In ***Appendix D, “Comparison with Weight Modification Method”***, we compare Safe-CB with a weight-modification approach adapted from state-of-the-art hallucination suppression techniques for multimodal models.\
> > > In ***Appendix G, “Direct supervised training of the model codebook”***, we discuss and compare the performance of directly ***supervised training on the codebook***, with experimental results shown in ***Table 11***.\
> > > Under these various baselines, our method achieves superior performance in both image safety and visual quality.
> > >
> > > &nbsp;&nbsp;&nbsp;&nbsp;**(2)** Our method for constructing the safe codebook can also be regarded as a special fine-tuning process. Unlike conventional approaches that train the entire model network or use LoRA, ***we only optimize a single codebook***.
> > >
> > > &nbsp;&nbsp;&nbsp;&nbsp;**(3)** To follow the reviewer’s suggestion, we have ***added new baselines including direct supervised fine-tuning (SFT) and DPO model training***. The experimental results are as follows:
> > > | Method | &nbsp;&nbsp; Sexual &nbsp;&nbsp; | &nbsp;&nbsp; Violence &nbsp;&nbsp; | &nbsp;&nbsp; Self-harm &nbsp;&nbsp; | &nbsp;&nbsp; Shocking &nbsp;&nbsp; | &nbsp;&nbsp;&nbsp;&nbsp;&nbsp; FID &nbsp;&nbsp;&nbsp;&nbsp;&nbsp; |
> > > | :--- | :---: | :---: | :---: | :---: | :---: |
> > > | **Baseline** | 0.1278 | 0.3981 | 0.4019 | 0.5327 | 68.83 |
> > > | **SFT** | 0.1062 | 0.2948 | 0.3271 | 0.4594 | 79.01 |
> > > | **DPO** | 0.1098 | 0.3233 | 0.3348 | 0.4710 | 75.42 |
> > > | **Safe-CB** | 0.0440 | 0.2106 | 0.2275 | 0.3371 | 70.66 |
> > >
> > > &nbsp;&nbsp;&nbsp;&nbsp;**① Experiment setting:** For these experiments, we used 2,000 harmful/safe prompt pairs from the CoPro dataset [1,2] to generate corresponding images.In the SFT training stage, we trained on pairs consisting of harmful prompts and safe images, with a learning rate of 5e-6 for 3 epochs. In the DPO training stage, we constructed triplet training data using harmful prompts, harmful images generated from harmful prompts, and safe images generated from rewritten safe prompts. DPO training stage with  learning rate=2e-6 and 𝛽=0.1, also for 3 epochs. All experiments are based on the Janus unified model.
> > >
> > > &nbsp;&nbsp;&nbsp;&nbsp;**② Analysis1:** Results show that training methods that directly guide the model during the next-token discrete image generation process do not achieve satisfactory performance, especially when learning abstract safety constraints. This is because image tokens are essentially local discrete representations without explicit semantics, making it difficult for the model to learn effective mapping logic between these abstract sequences and fine-grained safety constraints.
> > >
> > > &nbsp;&nbsp;&nbsp;&nbsp;**③ Analysis2:** Compared with training-based methods, our safe codebook also ***requires much less data***. As shown in Section 4.4 of the main paper, our Safe-CB already achieves strong safety performance using only 300 harmful/safe data pairs.
> > >
> > > [1] Latent guard: a safety framework for text-to-image generation. ECCV 2024.\
> > > [2] Align Guard: Scalable Safety Alignment for Text-to-Image Generation. ICCV 2025.
> > >
> > > **3. Regarding the impact on generation quality (e.g., Aesthetic Score, PickScore, HSPv2, etc.)**, we have further provided evaluations on these metrics, and the results are as follows:
> > > | Method | Aesthetic score | &nbsp;&nbsp;&nbsp;&nbsp; PickScore &nbsp;&nbsp;&nbsp;&nbsp; | &nbsp;&nbsp;&nbsp;&nbsp;&nbsp;&nbsp; HSPV2 &nbsp;&nbsp;&nbsp;&nbsp;&nbsp;&nbsp; |
> > > | :--- | :---: | :---: | :---: |
> > > | **Baseline** | 4.2290 $\pm$ 0.341 | 20.77 $\pm$ 0.51 | 27.11 $\pm$ 0.42 |
> > > | **Safe-CB** | 4.2308 $\pm$ 0.376 | 20.53 $\pm$ 0.78 | 27.07 $\pm$ 0.71 |
> > >
> > > As indicated by the experimental metrics, the visual quality of images generated by our model does not degrade significantly after applying our method.
> > >
> > > **Finally, Thanks to the reviewer again for valuable suggestions, which significantly helped us improve the quality and clarity of the paper.**

---

### Decision · Program_Chairs · 2026-04-30

**Decision:**

Accept (regular)

**Comment:**

One of the major issues raised by Reviewer Z9NC is the lack of a strong baseline for supervised fine-tuning or RLHF in its application. In particular, the proposed method for updating the codebook is required; a broader comparison would be persuasive. Although the authors argued that there is no prior work on 'safety in unified autoregressive image generation', there are multiple prior works on 'safety in image generation'. The reviewer argued that it would be worth comparing with previous representative safety methods in autoregressive image generation to support the claim that it is critical. AC shared the view with Reviewer Z9NC for this matter.

However, also noted that the experiments in Appendices D and G partially support this issue by modifying the model codebook's weights or providing direct supervision, which mitigates concerns. AC recommended proactively comparing other safety methods across general image generation models to support the claim and provide readers with a reference.

In light of this concern, the authors are encouraged to consider additional relevant references on safe image generation, such as the DPO-based approach using synthetic positive and negative images [a], and a training-free text embedding guidance method [b].

Throughout the rebuttal, other major issues are effectively discussed, and the assessment leans towards the positive or maintains positive scores. Therefore, AC recommends acceptance. Please revise the manuscript as requested by the reviewers, including the placeholder running title at the top of the pages.

--

[a] Direct Unlearning Optimization for Robust and Safe Text-to-Image Models (Park et al., NeurIPS 2024)

[b] Training-Free Safe Text Embedding Guidance for Text-to-Image Diffusion Models (Na et al., NeurIPS 2025)